# Utilization of Polymeric Micelles as a Lucrative Platform for Efficient Brain Deposition of Olanzapine as an Antischizophrenic Drug via Intranasal Delivery

**DOI:** 10.3390/ph15020249

**Published:** 2022-02-18

**Authors:** Hadel A. Abo El-Enin, Marwa F. Ahmed, Ibrahim A. Naguib, Shaymaa W. El-Far, Mohammed M. Ghoneim, Izzeddin Alsalahat, Hend Mohamed Abdel-Bar

**Affiliations:** 1Department of Pharmaceutics and Industrial Pharmacy, College of Pharmacy, Taif University, Taif 21944, Saudi Arabia; 2Department of Pharmaceutical Chemistry, College of Pharmacy, Taif University, Taif 21944, Saudi Arabia; marwa.farg@tu.edu.sa (M.F.A.); i.abdelaal@tu.edu.sa (I.A.N.); 3Division of Pharmaceutical Microbiology, Department of Pharmaceutics and Industrial Pharmacy, College of Pharmacy, Taif University, Taif 21944, Saudi Arabia; shfar@tu.edu.sa; 4Department of Pharmacy Practice, College of Pharmacy, AlMaarefa University, Riyadh 13713, Saudi Arabia; mghoneim@mcst.edu.sa; 5UK Dementia Research Institute Cardiff, School of Medicine, Cardiff University, Cardiff CF24 1TP, UK; alsalahati@cardiff.ac.uk; 6Department of Pharmaceutics, Faculty of Pharmacy, University of Sadat City, Sadat City 32897, Egypt; hend.abdelbar@fop.usc.edu.eg

**Keywords:** antischizophrenic drug, nanocarriers, olanzapine, nose to brain, induced schizophrenia-like behavior

## Abstract

Schizophrenia is a mental disorder characterized by alterations in cognition, behavior and emotions. Oral olanzapine (OZ) administration is extensively metabolized (~up to 40% of the administrated dose). In addition, OZ is a P-glycoproteins substrate that impairs the blood–brain barrier (BBB) permeability. To direct OZ to the brain and to minimize its systemic side effects, the nasal pathway is recommended. OZ-loaded polymeric micelles nano-carriers were developed using suitable biodegradable excipients. The developed micelles were physicochemically investigated to assess their appropriateness for intranasal delivery and the potential of these carriers for OZ brain targeting. The selected formula will be examined in vivo for improving the anti-schizophrenic effects on a schizophrenia rat model. The binary mixture of P123/P407 has a low CMC (0.001326% *w*/*v*), which helps in maintaining the formed micelles’ stability upon dilution. The combination effect of P123, P407 and TPGS led to a decrease in micelle size, ranging between 37.5–47.55 nm and an increase in the EE% (ranging between 68.22–86.84%). The selected OZ–PM shows great stability expressed by a suitable negative charge zeta potential value (−15.11 ± 1.35 mV) and scattered non-aggregated spherical particles with a particle size range of 30–40 nm. OZ–PM maintains sustained drug release at the application site with no nasal cytotoxicity. In vivo administration of the selected OZ–PM formula reveals improved CNS targeting and anti-schizophrenia-related deficits after OZ nasal administration. Therefore, OZ–PM provided safe direct nose-to-brain transport of OZ after nasal administration with an efficient anti-schizophrenic effect.

## 1. Introduction

Schizophrenia is a mental disorder characterized by hallucinations, alterations in cognition, behavior, emotion, and disorganized speech that severely cause distress and daily functional impairment [1,2]. Approximately 23 million people worldwide are suffering from schizophrenia, 1% of whom are in adolescence or early adulthood [3,4]. Moreover, the early mortality of schizophrenia patients is still high regarding patients’ age [5,6].

Olanzapine (OZ) is a typical novel antipsychotic agent approved by the FDA for the management of both positive and negative symptoms of schizophrenia and bipolar disorders caused by moderate to severe mania [7,8]. OZ selectively binds to two primary central targeting receptors, namely dopamine (D2) and serotonin (5-HT2c). In addition, it can target the muscarine cholinergic receptors M1–M5, adrenergic receptors alpha 1, and histamine H1 receptor [9,10]. OZ is known as 2-methyl-4-(4-methyl-1-piperazinyl)-10H-thieno(2,3-b) benzodiazepine with a partition coefficient (Log p) of 2.97 [11]. OZ is classified in the class II category under the biopharmaceutical classification system (BCS) with aqueous solubility of 0.24 mg/mL and positive BBB permeability at 0.9652 probabilities [12]. Orally administered OZ is extensively hepatically metabolized (~up to 40% of the administrated dose) to form a large inactive metabolic fraction [13]. Continuous OZ oral administration has persuaded many metabolic adverse effects, such as type 2 diabetes, insulin resistance (hyperinsulinemia), obesity, fatty liver and dyslipidemia [14,15]. In addition, the induced P-glycoproteins efflux by orally administered OZ was found to impair blood–brain barrier (BBB) permeability [13,16,17].

To direct OZ to the brain through the bypass of BBB, and to minimize its systemic side effects, the nasal pathway is recommended [18,19,20]. Nasal mucosa (in the nasal cavity) is a highly permeable membrane that is widely used as a non-invasive approach to bypass the BBB [18]. Olfactory neurons in the olfactory epithelium direct the anatomical olfactory information into the brain [19,21]. Nanotechnology is a promising approach to overcome the nasal mucosal barrier to achieve optimum OZ concentrations in the brain.

Polymeric micelles (PM) are kinetically stable spontaneous amphiphilic block copolymer nanocarriers in aqueous dispersions above the critical micelle concentration (CMC) [22,23]. These nanoparticles have a hydrophobic core–hydrophilic chain that mediates the loading of hydrophobic drugs into the core. PMs are preferred over other nanocarriers for brain delivery, as the particle size is ideal for passing the BBB and improving the encapsulation efficiency (EE%) [24,25]. Moreover, PM proved to increase the hydrophobic drugs’ solubility, stability and bioavailability [26,27].

Development of nasal OZ-loaded PM using biodegradable and biocompatible permeation enhancers is supposed to avoid the damage of the nasal mucosa. TPGS, d-α-Tocopheryl polyethylene glycol 1000 succinate, was investigated as a biocompatible and biodegradable non-ionic surfactant. Additionally, it enhances nanocarrier permeation and cellular uptake [28,29]. Furthermore, TPGS is a P-glycoprotein (P-gp) inhibitor [30], which could improve olanzapine BBB permeability.

Polymeric micelles comprising pluronic 123 (P123) and TPGS have the potential to improve cargo transport across the BBB by inhibiting multidrug transporters such as P-glycoprotein (P-gp). Furthermore, poloxamers, as Kolliphor^®^P407 (P407), have the ability to enhance the hydrophobic drug dissolution rate even in distilled water [31,32]. In addition, the pluronic copolymer tri-block structure of polyethylene oxide (PEO) and polypropylene oxide (PPO) have the ability to increase drug retention times [33,34].

Herein, this study aimed to develop OZ–PM using TPGS, P P123 and P407 for nose-to-brain targeting of OZ. The prepared micelles were physicochemically characterized to assess their appropriateness for intranasal delivery, and the potential of these carriers for brain targeting of OZ as an antischizophrenic agent in rats was investigated. The selected formula was examined for improving the antischizophrenic effects on a schizophrenia rat model using induced schizophrenia-like behavior.

## 2. Results and Discussion

### 2.1. Critical Micelle Concentration

In aqueous solutions, pluronic can self-assemble into micelles at a concentration above the CMC. CMC influences micelles’ in vitro and in vivo stability. The CMC of P123/P407 binary mixture was determined using iodine UV spectroscopy method. The data presented in Figure 1 revealed that CMC of P123/P407 is 0.001326% *w*/*v*. The low P123/P407 CMC value benefits in maintaining the formed micelles’ stability upon dilution. The latter indicates their high stability and integrity when extremely diluted in the body [35].

### 2.2. Thin Film Formation

Pluronic (P123/P407) and TPGS mixture at a concentration above CMC was self-assembled to prepare the PM, where the central core of these micelles are composed of a hydrophobic block (P123/P407 binary mixture) and an outer shell is formed from a hydrophilic block (TPGS). Such a structure improves the encapsulated hydrophobic drugs’ incorporation, stability, solubility and retention time, which could improve its absorption rate and bioavailability [36].

### 2.3. Optimization of OZ Polymeric Micelles Using D-Optimal Design

In general, nanocarriers’ physicochemical properties influence their intracellular uptake and subsequent therapeutic effects [37]. The optimization of different OZ–PM was conducted using D-Optimal design to affirm the influence and interactions of different critical process parameters (CPPs) on critical quality attributes (CQAs), namely particle size expressed as average hydrodynamic diameter (Y1) and EE% (Y2) (Table 1). The design generated 16 formulae, and the investigated CPPs were: (A) P123 (60–70%), (B) P407 (20–30%), (C) TPGS (5–10%).

A polynomial equation was used to express the relation between the variable CPPs and each of the stated CQAs. After excluding non-significant factors, the statistical models with the greatest R^2^ values (adjusted and predicted), with a difference of less than 0.2, and the lowest PRESS value were chosen [38]. The high R^2^ and the low PRESS values ensure the closeness of the predicted and experimental results as well as the ability of the developed model to predict the experimental results. The signal-to-noise ratio, defined as adequate precision, was more than four, demonstrating the models’ ability to span the design space [39].

Accordingly, the quadratic model was chosen as the best fit statistical model, (Appendix A). The positive regression coefficient sign implies a direct relationship between CPPs and CQAs, whereas the negative sign implies an inverse relationship [40]. Three-dimensional-response surfaces are graphical illustrations, indicating the relationship between two CPPs while all other independent factors are kept constant.

### 2.4. Influence of Different Critical Process Variables on Particle Size (Y1)

Table 1 shows that the fabricated OZ–PMs had a particle size (expressed as average hydrodynamic diameter) in the range of 37.5–47.55 nm. All formulae showed a PDI value of less than 0.2 with a unimodal distribution. Effect of different significant CPPs on particle size is presented according to the following equation:(1)$$\mathrm{Particle}\;\mathrm{size}\;\left(Y1\right)=+49.91\;A+42.96\;B+88.44\;C-30.24\;\mathrm{AB}-68.56\;\mathrm{AC}-80.96\;\mathrm{BC}$$

The data in Appendix A and the associated ANOVA findings (Appendix A) show that the regression coefficients of the tested CPPs all had *p* values < 0.05, suggesting that they have a significant influence on PM particle size.

From Equation (1), the increase in each polymer concentration leads to an increase in the micelle size, which is related to the increase in the hydrophobic core. From the data listed in Table 1 and Figure 2A and Figure 3A, it can be deduced that the interaction effect of P123, P407 and TPGS mixture had a negative effect on the OZ–PM size. As previously reported, the pluronic block copolymers have a significant effect on micelle size, as it contains polyethylene oxide (PEO) and polypropylene oxide (PPO) blocks [41]. The high PPO segment length of P123 and the higher hydrophobicity of mixed micelles reduced the micelle size [42].

Effect of variable critical process parameters on OZ EE% (Y2):

Table 1 indicates that the fabricated OZ–PMs EE% ranged between 68.22–86.84%. The effect of various significant CPPs on EE% was described according to the following equation:(2)$$\mathrm{EE}\%\;\left(Y2\right)=+105.52\;A+93.43\;B-5.04\;C-74.19\;\mathrm{AB}+42.05\;\mathrm{AC}+67.93\;\mathrm{BC}\;$$

All the examined CPPs had a significant effect on the OZ EE% with *p* values <0.05 as represented in Appendix A and the corresponding ANOVA results represented in Appendix A. According to Equation (2), it was found that increasing the P123 and P407 concentration had a significant effect on increasing the OZ EE%, while increasing TPGS led to a significant decrease in EE%. The hydrophobic properties of both polymers and their higher PPO/PEO ratio improve OZ solubilization efficiency and consequently improve EE%. The binary coefficient of both polymers was less than each polymer alone and had less effect on EE%. This was in agreement with previously reported results by Thanitwatthanasak et al., 2019 [43]. The formation of stable reactions between the TPGS’ aromatic ring and the PPO group of pluronics (P123/TPGS or P407/TPGS), and/or encapsulated OZ could be attributed to the increase in the EE% coefficient [44].

### 2.5. Design Space and Optimization

Design space is a multidimensional combination of interactions of CPPs that were identified through design of experiment (DoE) data and verified to afford QTPP. It was constructed by superimposing the impacts of several CPPs on CQA contour graphs. The yellow zone shows CPP values that have been optimized to satisfy QTPP criteria, namely minimum particle size and maximum EE percentage (Figure 4 and Figure 5).

Based on the high desirability (=0.812), one formula (OZ–PMs) was chosen as a benchmark to verify the constructed models. Table 2 illustrates its composition and the relevant predicted and experimental particle size and EE% values. Depending on the low values of predicted error %, the developed model was satisfying for investigating and predicting the CPPs for OZ–PM preparation with the required QTPP.

### 2.6. Characterization of the Prepared OZ–PMs

The prepared olanzapine PMs had a zeta potential value of −15.11 ± 1.35 mV. The negative charge could be attributed to the nature of the components. The drug loading efficiency (DL %) was 8.52 ± 0.58%. The used polymers were naturally nonionic; therefore, the negative charge was related to the TPGS and/or entrapped OZ. The negative charge improves the micelle stability by preventing their aggregation and improves the nasal absorption through the olfactory pathway by transcytosis [45]. Additionally, at low charge (<−30 mV), lower repulsion was produced between the PM and membrane. Therefore, electrostatic repulsion force probably did not influence drug penetration [46,47].

The transmission electron micrograph of OZ–PMs as represented in Figure 6 showed scattered non-aggregated spherical particles with a particle size range of 30–40 nm, which is in consistency with the particle size obtained by DLS. The spherical nano-size OZ–PMs could enhance drug permeability and improve its retention effects [48,49].

### 2.7. In Vitro Release

The OZ in vitro release from the OZ–PM was tested in PBS pH 7.4 containing 0.5% Tween 80 to maintain the sink condition using a dialysis method. As noticed from Figure 7A, the accumulative drug released from the PM was able to sustain OZ release for 24 h with no burst release. This result was in agreement with a previously reported study where TPGS could increase the PM hydrophobicity by improving the polymer chain interactions in the micelle’s core [50], which strengthen the drug and micellar system interactions and thus slow OZ release [51]. By applying different kinetic equations, the best fitted model of the in vitro OZ release data followed the Higuchi diffusion kinetics model (Figure 7B).

### 2.8. In Vitro Cytotoxicity

The in vitro cytotoxicity test was conducted using MTT assay on Calu-3 cells after incubation with OZ–PM or empty micelles for 48 h (Figure 8). The IC_50_ of plain PMs and OZ–PMs was not decisively detected (>300 µM), indicating the modest or even possible lack of cytotoxicity from both formulations after incubation for 48 h. This could be attributed to the presence of pluronics and TPGS in the PMs, as they are biodegradable and biocompatible polymers [52,53].

### 2.9. In Vivo Biodistribution Study

Figure 9 demonstrates the OZ concentrations in rats’ plasma and brains after administration of different treatments against time and the pharmacokinetic variables as well as DTE (%) and DTP as represented in Table 3. It was noted that the brain C_max_ and AUC_(0–480)_ after administration of intranasal OZ–PM were significantly higher than IV OZ solution. The increase in drug duration at the nasal site was indicated by a high MRT value. The shorter Ke value (0.19 h^−1^) in the brain following intranasal OZ–PM administration and the extended MRT in addition to the increase in its AUC were expected for PM formulations as a result of protecting the drug inside their core. Additionally, the micelle nanosize could be attributed to the higher drug concentration in the brain, as the nanosize facilitates deeper transport of drug particles into the olfactory epithelial cell layers [54,55].

The previous result could also be proved by the high DTE% and DTP% values. The latter indicates that the drug could be transported directly by the axonal CNS drug transport pathway rather than by the systemic pathway [56]. Additionally, TPGS could increase the nasal absorption through inhibiting the P-gp brain efflux effect of the drug. They could also increase the nasal absorption in the epithelial cells of the nasal mucosa by reducing the draining effect [57]. In addition, the used pluronics (P123/P407) have the ability to overcome the P-gp efflux mechanism and to increase drug solubilization, thereby enhancing its cellular uptake [58].

### 2.10. Histopathological Examinations of the Nasal Mucosae

Figure 10 demonstrates the light photomicrographs of the nasal mucosa of rats exposed to OZ–PM for 14 days, compared to control animals (with untreated nasal mucosa). No damage signs were detected on the nasal mucosa after OZ–PM administration. No alterations were detected in the epithelial lining mucosa or in the cartilaginous layer. The integrity of the lamina propria and the vessel was also maintained. These results indicate that the OZ–PM formulation with its components can be safely applied nasally.

### 2.11. Paw Test

The paw test was conducted to estimate the drug’s capacity to prevent the fore and hindlimb spontaneous withdrawal. The antipsychotic potential effects were linked to an increase in hindlimb retraction time (HRT), while the possibility for extrapyramidal side effects was associated with a rise in forelimb retraction time (FRT) [59]. As represented in Table 4, it was also observed that nasal administration of OZ–PM showed significantly higher HRT and FRT values as compared to control group. The increase in HRT implies that a higher amount of OZ has reached the brain and indicates the brain targeting effect of nasal OZ–PM formulation (*p* < 0.001). By comparing HRT and FRT values after nasal administration compared to the IV solution, OZ–PM had a significantly higher antipsychotic with potentially lower extrapyramidal side effects (*p* < 0.001). These can be attributed to the controlled drug release from the formulation, which explains the absence of EPSs [60].

### 2.12. Open Field Test

The number of crossed squares in 60 min by rats is summarized in Figure 11. Analysis of locomotor activity revealed that there was a significant increase in the number of crossed squares after a single ketamine dose (25 mg/kg) (*p* < 0.001). Locomotor activity testing was used as a modeling of the positive symptoms of schizophrenia [61]. Schizophrenia is described as a severe and chronic disorder of the brain that can be attributed to abnormalities in the levels of neurotransmitters. The ketamine dose affects the animals’ behavior as a result of its effect on the neurotransmitters.

Ketamine-induced hyperlocomotion was produced by a selective increase in glutamate and dopamine release in the prefrontal cortex, nucleus accumbens and striatum [62,63]. Pretreated rats with OZ solution or OZ–PM with a dose of 2 mg/kg showed a significant decrease in the ketamine-induced hypermobility, which was indicated by decreases in the observed rats’ mobility (*p* < 0.001). The significant reduction in the crossing squares number was due to the ability of OZ to block the 5-HT2A receptors on the glutamatergic terminals that cause the spontaneous reduction in the ketamine-induced stimulation effect in the dopamine–glutamate pathway [64,65]. The higher reduction in the animals’ mobility after nasal administration of OZ–PM compared to the IV–OZ solution was related to the higher OZ concentration in the brain.

## 3. Materials and Methods

### 3.1. Materials

Pluronic^®^P123 (P123), Kolliphor^®^P407 (P407), Tween 80,d-α-tocopheryl polyethylene glycol 1000 succinate (TPGS) and Kolliphor^®^TPGS were supplied by Sigma-Aldrich (St. Louis, MO, USA). Olanzapine was purchased from Sigma-Aldrich (St. Louis, MO, USA). Disodium hydrogen phosphate, glacial acetic acid, methanol, ethanol and sodium hydroxide (analytical grade) were provided by El-Nasr Chemical Co (Cairo, Egypt).

### 3.2. Critical Micelle Concentration (CMC) Determination

Aqueous solutions of polymers P123 and P407 with different concentrations at a constant concentration of TPGS (1% *w*/*w*) were prepared. CMC was calculated using the iodine UV spectroscopy method [66]. To each of the P123/P407 binary mixture solution, 25 μL of KI/I_2_ standard solution was added. The KI/I_2_ standard solution was prepared by dispersing 0.5 g iodine and 1 g of potassium iodide in 50 mL of deionized water. The mixtures were incubated in the dark at 25 °C for 12 h. The absorbance was measured at 366 nm, and the average of three replicates was plotted against the logarithm of polymer concentration. The observed point with a sharp increase in absorbance corresponding to the concentration of the polymer was considered as the CMC value.

### 3.3. Thin Film Formation

The OZ–PMs were prepared using a thin-film hydration method. In brief, OZ and non-ionic surfactant (P407, P123 and TPGS) mixtures in different ratios were dissolved in a round-bottom flask using 10 mL ethanol. The dissolving medium (ethanol) was vaporized using a rotary vacuum evaporator at 50 °C under low pressure. The system was revolved at 150 rpm for 40 min until the formation of thin dry film on the inner flask’s wall. The thin film was further dried under vacuum at room temperature to ensure complete film drying and removal of residual ethanol (if any) for 24 h. To create micelles, the dry film was moistened with 5 mL of distilled deionized water and vigorously shaken for 3 min. The produced micelles were centrifuged at 10,000× *g* rpm for 10 min, and the supernatant was filtered through a 0.45 μm membrane filter to remove unincorporated OZ aggregates. Empty micelles (without OZ) were also prepared using the same steps.

To optimize OZ–PM particle size and EE% for improved nasal absorption, D-Optimal design was involved and is shown in Table 5.

### 3.4. Evaluation of the Prepared OZ-Micelles

#### 3.4.1. Particle Size and Zeta Potential of OZ–PM Micelles Analysis

The mean micelles size (expressed as average hydrodynamic diameter), polydispersity index (PI) and zeta potential value of OZ–PM micelles were measured at 25 °C with a detection angle of 90° using a dynamic light scattering technique (NanoBrook 90 Plus PALS, Brookhaven Instruments, Holtsville, NY, USA). Each sample was diluted 10 times using de-ionized water (avoiding multi-scattering phenomena). The data collected were the means of three runs ± standard deviation (SD).

#### 3.4.2. Drug Entrapment Efficiency (EE%) and Drug Loading (DL %) Percentages Determination

Drug entrapment efficiency was determined in the prepared OZ–PM micelles using a direct method. Each sample was dissolved in methanol/water (70/30) to disrupt the polymeric micelle, and the OZ amount was measured spectrophotometrically (Shimadzu, model UV-1601 PC, Kyoto, Japan) at λ_max_ 272 nm following sufficient methanol dilution of the clear supernatant [66]. Both EE% and DL% were determined in triplicate using the following equations:(3)$$EE\%=\frac{Weight\;of\;OZ\;loaded}{Weight\;of\;OZ\;intially\;added}\times 100$$
(4)$$LC\%=\frac{Weight\;of\;OZ\;loaded}{Weight\;of\;micelles}\times 100\;$$

#### 3.4.3. Morphological Examination

Transmission electron microscope (TEM) (Jeol JEM 1230, Tokyo, Japan) was used to determine the OZ–PM morphological aspects. On a copper grid, one drop of the selected OZ–PM dispersion was deposited. A filter paper was used to remove the excess. After that, one drop of phosphotungstic acid aqueous solution (2% *w*/*v*, negative staining) was applied, and the excess was disposed of in the same manner. Finally, TEM was used to study the grid.

### 3.5. In Vitro Release

To determine the in vitro release rate of OZ from the prepared selected OZ–PM, 2 mg was suspended in 0.5 mL of PBS pH 7.4 with gentle vortexing and was placed in a dialysis membrane (12,000–14,000 molecular weight cut-off, Spectrum Laboratories Inc, Piscataway, NJ, USA). The closed bag was immersed in 100 mL of the released media (PBS with 0.5% Tween 80, pH 7.4) in a glass beaker. The system was placed on a platform shaker at 50 rpm for 24 h using a horizontal orbital shaker and incubated at 37 °C. At different time intervals, samples were withdrawn and replaced with fresh buffer during the process. The amount of OZ released was determined spectrophotometrically at λ_max_ 272 nm after appropriate dilution. The same steps were repeated with empty micelles to be used as a blank and compared to the release pattern of pure drug release separately.

To determine the release kinetics order, the drug release patterns were fitted to zero order, first order and Higuchi diffusion models [67]. The time required for 50% of the loaded medication to be released (T_50_%) was computed and statistically compared to the OZ release profile.

### 3.6. In Vitro Cytotoxicity

In vitro, OZ–PM cytotoxicity, compared to the drug free micelles, was tested by measuring membrane damage on Calu-3 cells using the MTT assay [68]. Calu-3 cells lines were supplied from the American Type Culture Collection (ATCC, Manassas, VA, USA). Calu-3 cells were cultured in 96-well plates at a density of 6 × 10^4^/well for 24 h under 5% CO_2_, 90% relative humidity, and at 37 °C. Different concentrations of OZ–PM or empty micelles were added to Calu-3 cells and incubated for 48 h. Consequently, 20 µL of 5 mg/mL of MTT solution was added to the cultured media and incubated for 4 h at 37 °C. Cell toxicity was analyzed by measuring absorbance at 570 nm with Synergy 2 Multi-Detection Microplate Reader, BioTek Instruments Inc. This assay was replicated three times for each experiment. The inhibitory concentration (50%) was determined, and the results were expressed as mean ± standard deviation. The negative control was prepared by adding culture medium alone (100% proliferation).

### 3.7. In Vivo Examination

The brain targeting effectiveness and the behavior study on schizophrenic rats of the selected OZ–PM were carried out in accordance with the recommendations of the National Institutes of Health Guide for the Care and Use of Laboratory Animals (NIH Publications No. 8023, revised 1978) with a protocol approved by the ethical committee of Al-Taif University, KSA; approval number: 43-022. Male Wistar rats weighing 180–220 g were caged and given free access to water and regular food at room temperature for one week before testing for acclimatization.

#### 3.7.1. Nasal Biodistribution Pattern Study

In this study, the rats were divided into two groups as follows: group 1, the positive control, in which rats were intravenously given OZ solution in phosphate buffer (pH 7.4) through their peripheral tail veins; group 2 received the selected OZ–PM formula after dispersion in PBS pH 7.4. OZ–PM was administered nasally using a micropipette (2–20 μL) into the nostrils of conscious rats attached to a low-density polyethylene microtip. The rats were allowed to inhale all the preparations gently. All animals were administrated OZ at a dose of 2 mg/kg [69].

Immediately after that, the rats were anaesthetized by inhalation of diethyl ether. Blood samples were withdrawn at different time intervals after dose administration. Blood samples were gathered from rats via retro-orbital venous plexus into heparinized test tubes and then centrifuged at 3000× *g* rpm at 4 °C for 15 min, and the supernatant (plasma) was collected and stored at −20 °C.

Animals were decapitated, skulls were opened, and brains were meticulously removed at the same time intervals, directly after blood collection. The brain mass was weighed and then homogenized with distilled water at 3-fold the brain weight. Brain homogenate was cooled, centrifuged at 3000× *g* rpm and 4° for 15 min, and then the supernatant was isolated and stored at −80 °C [70].

The OZ amount in every sample was assayed using a liquid chromatography tandem mass spectrometry (LC-MS/MS) mass spectrometer (MDS Sciex, Foster City, California, USA) using API-3200 mass spectrometer with a Turbo ion spray TM set at 5500 V.

The used mobile phase was a mixture of 0.05 M ammonium acetate buffer and acetonitrile (30:70 *v*/*v*) using a flow rate of 1.5 mL/min. Quantities of 0.25 mL of plasma or brain samples were individually added to a volume of 0.75 mL of torsimide (internal standard) in acetonitrile solution. Collected samples were vortexed, centrifuged at 3000× *g* rpm for 10 min, and the obtained clear supernatant was injected into the LC-MS/MS system (Injection volume was 10 μL).

Detection was carried out using multiple reaction monitoring mode; m/z 313 precursor ion to the m/z 256 for OZ and m/z 348 precursor ion to the m/z 263 for an internal standard using Luna C_18_ (Phenomenex, Torrance, CA, USA) (5 × 4 mm; 5 μm particle size) column at 500 °C.

Pharmacokinetic parameters T_max_, C_max_, AUC_(0–480)_, and AUC_(0-∞)_, elimination constant Ke, and mean residence time (MRT) were calculated using the pharmacokinetic software PK Solver-Add Ins for Microsoft Excel 2007. The drug-targeting efficiency percentage (DTE %) and direct-transport percentage (DTP %) were calculated according to the following formulae:(5)$$\mathrm{DTE}\%=\left(\frac{\frac{AUC\;brain}{AUC\;blood}\;i.n}{\frac{AUC\;brain}{AUC\;blood}\;i.v}\right)\times 100$$
(6)$$\mathrm{DTP}\%=\left(\frac{B\;i.n-Bx}{B\;i.n}\right)\times 100$$
where *Bx = $\frac{B\;i.v}{P\;i.v}x\;P\;i.n;\;$ *Bx is the brain AUC fraction contributed by systemic circulation through the BBB following intranasal administration, and B i.v and B i.n are the AUC0–480 (brain) following intravenous administration and intranasal administration, respectively, while P i.v and P i.n. are the AUC0–480 (blood) following intravenous administration and intranasal administration, respectively [71,72].

#### 3.7.2. Histopathological Studies

To study the effect of the selected OZ–PM formulation on nasal mucosa integrity, six male Wister albino rats (weighing 180–220 g) received 20 µL of OZ–PM formula nasally once daily for 14 days. Rats were sacrificed, the rats’ noses were separated and nasal specimens were collected from the nasal septum of the epithelial cell membrane. Nasal specimens were immediately fixed in 10% formaldehyde solution for 24 h and then washed, dried, clarified and embedded in melted paraffin at 56 °C in a hot-air oven for 24 h. The paraffin blocks were sectioned at a thickness of 4–5 microns and stained with hematoxylin and eosin. For histological examination, the sections were examined using a binocular Olympus CX31 microscope (Motic BA210, Schertz, TX, USA) under 400× magnification [73].

### 3.8. Pharmacodynamics Studies

#### 3.8.1. Paw Test

The paw test was conducted by using a Perspex platform with dimensions of 30 × 30 cm and a height of 20 cm. At the top of the platform, two smaller holes were located for the forelimbs (4 cm diameter), two larger holes for the hindlimbs (5 cm diameter) and a slit for the tail [74]. The rats were distributed into 3 groups consisting of six each. In the negative control group, the solvent only was intravenously administered. Carefully, the rats’ hindlimbs were lowered in the holes, followed by forelimbs after 30 min after solvent administration. The forelimb retraction time (FRT) and the hindlimb retraction time (HRT) were recorded, where the FRT was the time taken by the rat to withdraw one forelimb, and the HRT was the time taken by the rat to withdraw one hind limb with considering 1 s as the minimum recorded time, while the maximum time was up to 30 s. The paw test was repeated twice: one after intravenous administration of OZ solution and the other after intranasal administration of selected OZ–PM formula [75].

#### 3.8.2. Schizophrenia Rat Model

Twenty-four rats were housed in six cages, six per cage. Group 1 was the negative control (C) group, where the rats were normally fed with a basal diet and injected intraperitoneally (i.p) with 0.2 mL saline every day for a week. Schizophrenia was induced in rats (eighteen rats) by injecting a dose of 25 mg/kg of ketamine intraperitoneally to each animal. Six rats were considered as the positive control group (group 2). Then, the rats were divided into two groups (group 3 and group 4), six each. Group 3 was injected intravenously with OZ–PB solution, while group 4 received a nasal daily dose (2 mg/kg) of OZ–PM. Group 3 and 4 were treated with a daily dose of 2 mg/kg for 7 days.

#### 3.8.3. Open Field Test

This test was performed on schizophrenic rats to estimate the rats’ locomotor activity. Each rat was located centrally in an open-field apparatus (40 × 40 × 30 cm; Accuscan Instruments, Columbus, OH). The apparatus was divided by black lines into 16 small squares (10 × 10 cm). The rats were moved to the testing room in their home cages and remained undisturbed for 2 h. Rats were gently handled by the base of their tails and placed individually into one of the four corners of the open field facing the center. The animal’s overall motor activity corresponding to walking was estimated by recording the ambulatory distance crossed by the rat (the total number of squares crossed) [76]. Obtained data were collected over a 60 min period.

## 4. Conclusions

The current study confirmed the feasibility of PM as a platform for the intranasal delivery of OZ in schizophrenic rats through in vitro and in vivo characterization. Herein, we have effectively tailored the PM composition to obtain an efficient platform for direct nose-to-brain transport of OZ. Different OZ-loaded PMs were fabricated by thin film hydration technique and optimized through D-Optimal DoE. The selected negatively charged OZ–PM with a particle size of 39.25 ± 2.35 nm and EE 82.15 ± 1.25% was able to control in vitro drug release up to 24 h with no burst effect. In vivo pharmacokinetics results demonstrated that the OZ–PM platform resulted in almost a four-fold increase in drug concentration in the brain compared to IV drug solutions. The targeting efficiency and direct nose-to-brain transport results confirmed the superiority of the intranasal route for efficient OZ–brain deposition. The current study also demonstrates the feasibility of the proposed PM to improve the therapeutic outcome of OZ in schizophrenic rats compared to IV drug solution. In addition, the OZ–PM showed less extrapyramidal side effects due to the lower systemic biodistribution than the OZ–IV solution.

## Figures and Tables

**Figure 1 pharmaceuticals-15-00249-f001:**
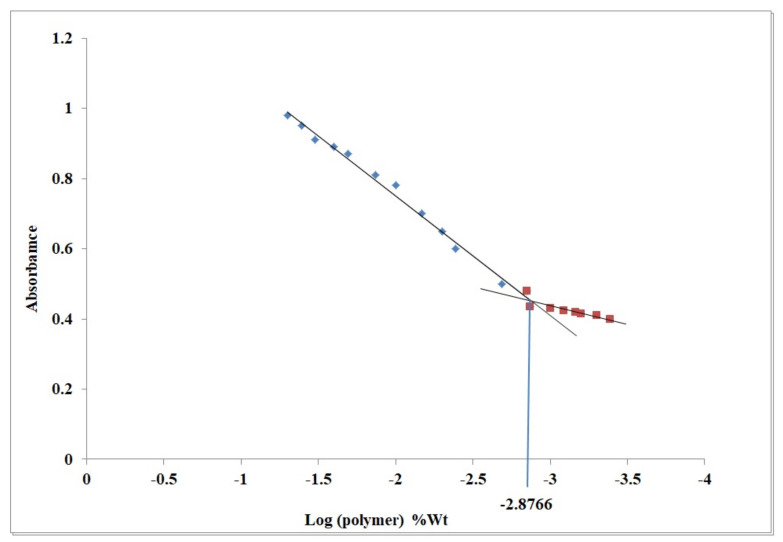
UV absorbance of I_2_ versus P123/P407 at different concentrations in water.

**Figure 2 pharmaceuticals-15-00249-f002:**
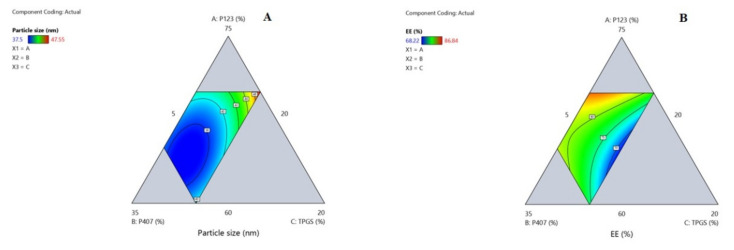
Contour plot for the interaction of different critical process parameters on OZ polymeric micelles’ particle size (**A**) and EE% (**B**).

**Figure 3 pharmaceuticals-15-00249-f003:**
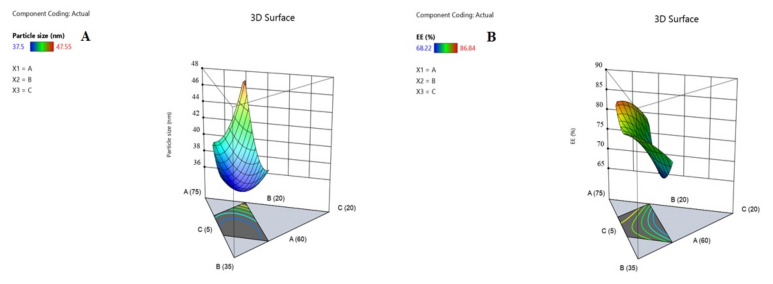
Three-dimensional surface plot for the interaction of different critical process parameters on OZ polymeric micelles’ particle size (**A**) and EE% (**B**).

**Figure 4 pharmaceuticals-15-00249-f004:**
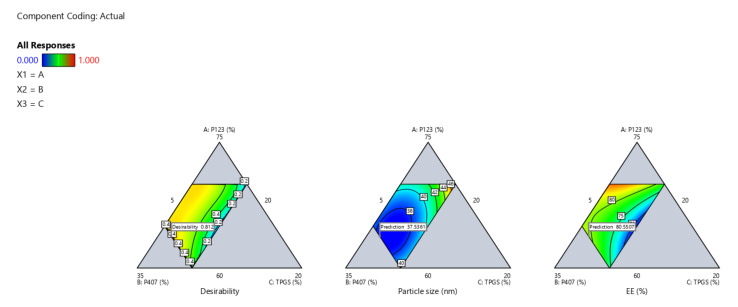
Contour plot for the OZ–PM desirability.

**Figure 5 pharmaceuticals-15-00249-f005:**
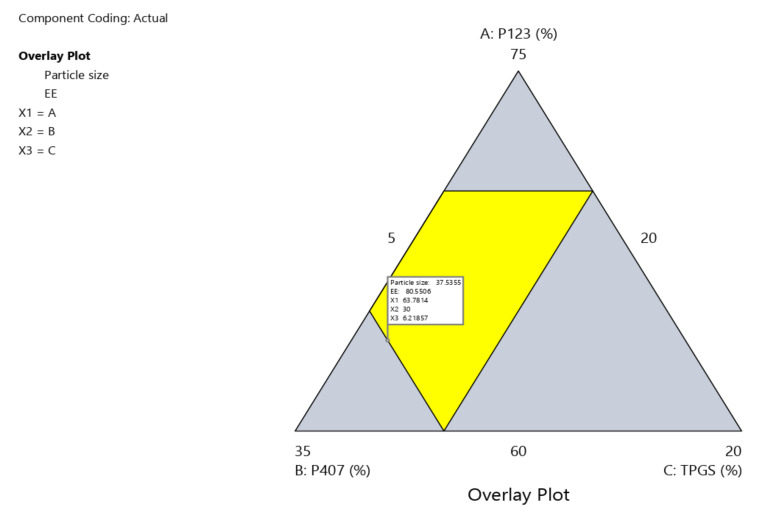
Design space region for the OZ–PMs.

**Figure 6 pharmaceuticals-15-00249-f006:**
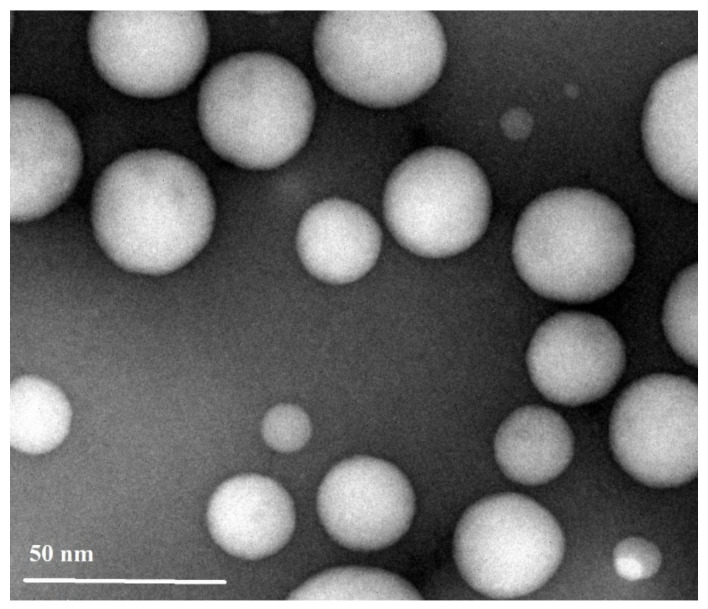
TEM of olanzapine polymeric micelles.

**Figure 7 pharmaceuticals-15-00249-f007:**
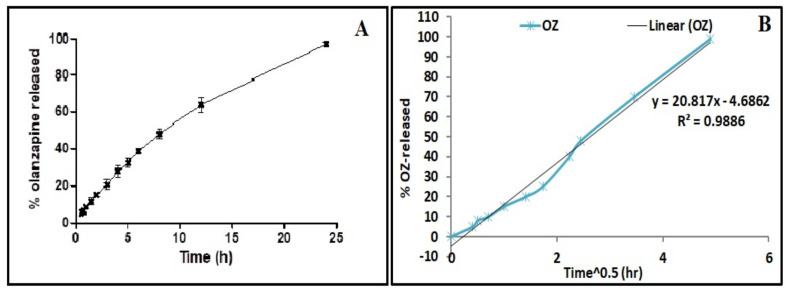
(**A**) In vitro release of olanzapine polymeric micelles in PBS pH 7.4. (**B**) Higushi equation for release kinetics of the selected OZ–PM formulae.

**Figure 8 pharmaceuticals-15-00249-f008:**
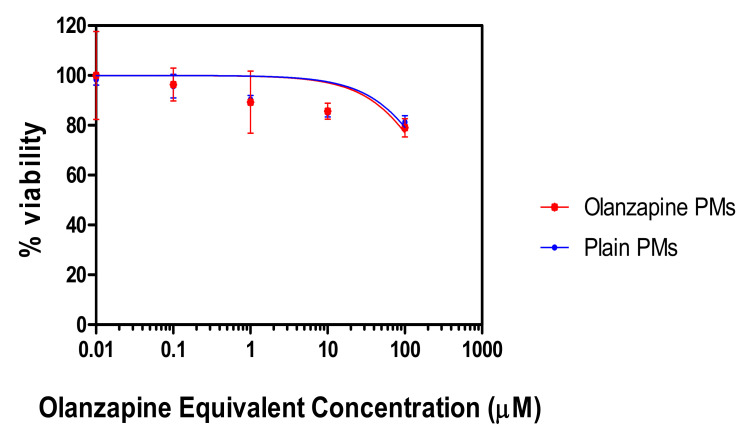
In vitro cytotoxicity of OZ polymeric micelles on Calu-3 cells.

**Figure 9 pharmaceuticals-15-00249-f009:**
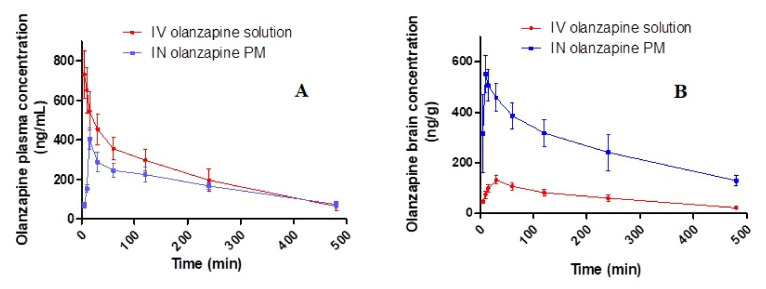
OZ concentrations in rats after administration of various formulations: (**A**) plasma concentrations; (**B**) brain concentrations.

**Figure 10 pharmaceuticals-15-00249-f010:**
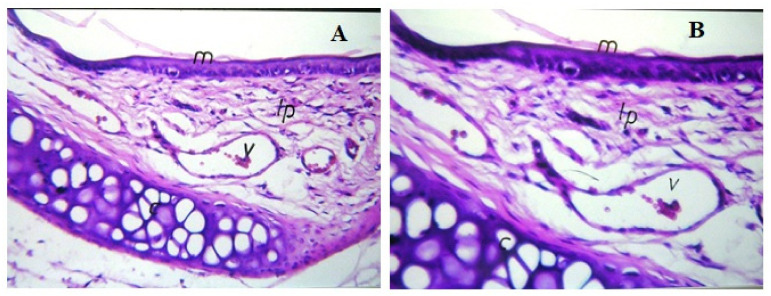
Light photomicrographs of the rat nasal mucosa following 14-day exposure to olanzapine PM (**A**) and control untreated animal (**B**) m, mucosa; v, vessel; lp, lamina propria; c, cartilaginous layer.

**Figure 11 pharmaceuticals-15-00249-f011:**
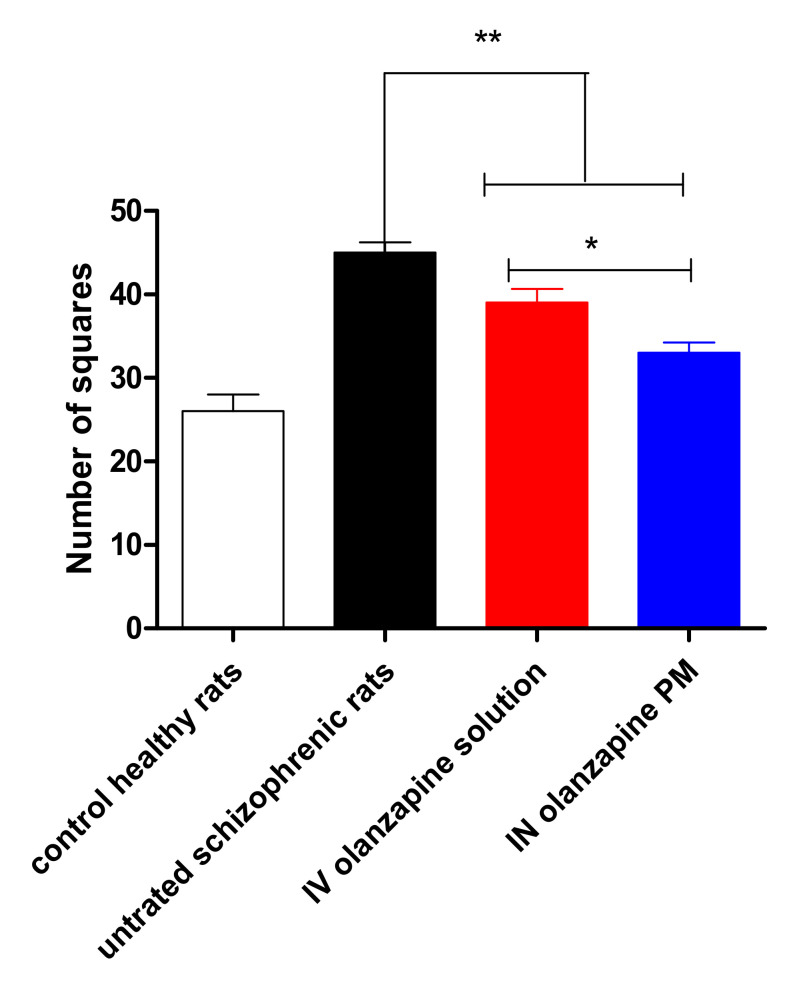
Assessment of pharmacodynamic effect of IV olanzapine solution and IN olanzapine PM in ketamine-induced schizophrenia in rats by open field test. *: *p*< 0.05; **: *p*< 0.001.

**Table 1 pharmaceuticals-15-00249-t001:** Experimental design matrix’s critical process parameters and the critical quality attributes related to them.

Run	A: P123 (%)	B: P407 (%)	C: TPGS (%)	Average Hydrodynamic Diameter (nm)	PDI	EE (%)
1	68.125	23.125	8.75	41.25	0.12	76.21
2	68.125	24.375	7.5	39.11	0.19	78.88
3	60	30	10	40	0.2	75.55
4	68.125	25.625	6.25	38.5	0.19	80.24
5	70	22.5	7.5	42.1	0.11	83.11
6	65	30	5	37.89	0.17	80.51
7	70	25	5	41	0.13	83.66
8	65	25	10	40.5	0.13	68.79
9	70	20	10	47.44	0.15	78.2
10	62.5	30	7.5	37.5	0.19	79.63
11	65	25	10	40.45	0.2	68.22
12	70	20	10	47.55	0.13	78
13	60	30	10	40.11	0.12	75.54
14	63.125	28.125	8.75	38.1	0.2	75.14
15	65	30	5	39.2	0.2	81.22
16	70	25	5	40.9	0.19	86.84

**Table 2 pharmaceuticals-15-00249-t002:** The experimental and predicted (A) particle size and (B) EE% of the optimized olanzapine PMs.

Parameter	P123 (%)	P407 (%)	TPGS (%)	Exp.	Pre.	% Pre. Error
Particle size (average hydrodynamic diameter) (nm) ^a,c^	63.78	30	6.22	39.25 ± 2.35	37.53	4.38
EE% ^b,c^				82.15 ± 1.25	80.55	1.94

^a^ Particle size was obtained by DLS. ^b^ Calculated as a percentage of the initial OZ added and determined by direct HPLC. ^c^ Expressed as mean ± SD (n = 3).

**Table 3 pharmaceuticals-15-00249-t003:** Pharmacokinetic parameters of intravenous olanzapine solution and intranasal polymeric micelles.

Parameter	In Plasma		In Brain	
	IV Olanzapine Solution	IN Olanzapine PM	IV Olanzapine Solution	IN Olanzapine PM
Cmax (ng/mL)		405.53 ± 54.15	132.31 ± 16.81	552.16 ± 29.61
Tmax (min)		15	30	15
AUC 0–480 min (ng/mL.h)	1776.21 ± 56.25	1359.37 ± 85.23	504.47 ± 38.52	2067.22 ± 105.36
AUC 0-∞ (ng/mL.h)	2022 ± 101.22	1740 ± 54.36	604.34 ± 45.36	2839.18 ± 50.36
MRT (h)	3.27 ± 0.25	4.11 ± 0.29	3.7 ± 0.24	4.41 ± 0.34
Kel (h-1)	0.26 ± 0.02	0.19± 0.01	0.215 ± 0.011	0.168 ± 0.014
Absolute bioavailability (F %)	100	76.53		
DTE (%)				535.93
DTP (%)				81.34

**Table 4 pharmaceuticals-15-00249-t004:** The pharmacodynamic assessment of IV olanzapine solution and IN olanzapine PM on normal rats.

Parameter	Control	IV Olanzapine Solution	IN Olanzapine PM
HRT (sec)	4 ± 1	11 ± 2	16 ± 3
FRT (sec)	6 ± 1	14 ± 2	9 ± 2

**Table 5 pharmaceuticals-15-00249-t005:** Critical process parameter levels, critical quality attributes and quality target product profile for OZ polymeric micelles were produced utilizing the D-Optimal design.

**CPPs**	**Levels**	
**(Coded Independent Variables)**	**Minimum**	**Maximum**
A: P123 (%) B: P407 (%) C: TPGS (%)	60 20 5	70 30 10
Type (mixture)	Total 100%	
CQAs (Responses)	QTPP (Constraints)	
Y1: Particle size (nm) Y2: EE (%)	Minimize Maximize	

## Data Availability

Data is contained within the article and Appendix A.

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
