# Peer review of "Utilization of Polymeric Micelles as a Lucrative Platform for Efficient Brain Deposition of Olanzapine as an Antischizophrenic Drug via Intranasal Delivery"

_pharmaceuticals, 2022, doi:10.3390/ph15020249_

Round 1

Reviewer 1 Report

The manuscript entitled “Utilization of Polymeric Micelles as a Lucrative Platform for Efficient Brain Deposition of Olanzapine as an Antischizophrenic Drug via Intranasal Delivery” deal with an interesting approach to treat schizoprhrenia through nose-to-brain delivery. The experimental data is well supported by in vivo studies, however I have some issues with some basic concepts, which should be clarified!

Please clarify the methodology of determination of CMC. Figure 1. shows the relation between polymer weight ratio and absorbance. To which polymer is related to the plotted weight ratio? Why the amount of TPGS was fixed to 1% w/w, however in the experimental design its concentration was investigated between 5-10% w/w?

The authors define Qulity by Design (QbD) elements (CPPs and QTTPs) based on the experimental design was constructed. How were these QbD elements selected? Only these parameters can have critical effect on the formulation?

In line 210 the authors claim the obtained zeta potential was negative, however a positive value is given. Please revise!

In line 213-216 the authors claim “The negative charge improves the micelle stability by preventing their aggregation and improves the nasal absorption of the drug by facilitating drug-mucous layer (positive charge) interaction and thus facilitating the drug diffusion through the epithelium layer”. This is an incorrect information, the nasal mucosa has negative surface charge, so there might be a repulsion between negatively charged nanoparticles and the mucosa, which may hinder drug absorption. The authors refer to Reference 54, however that paper does not deal with nasal absorption, please search for another suitable reference!

Table 2 shows the particle size of different composition. Which technique was applied for the determination of it? TEM or DLS? I guess DLS, but in that case it would be better to name as average hydrodynamic diameter. The PdI values are missing from the table, please add them!

How was TEM measurement conducted? Formulations were in liquid phase or solid phase during the measurement?

Why PBS pH=7.4 was applied for drug release study? Which circumstances was aimed to imitate? Nasal cavity, blood circulation, CSF of CNS?

Author Response

Dear / Editor-in-Chief, Pharmaceuticals journal

Dear/ Respected reviewers

Thank you for giving us the opportunity to submit a revised draft of our manuscript titled [Utilization of polymeric micelles as a lucrative platform for efficient brain deposition of olanzapine as an antischizophrenic drug via intranasal delivery], manuscript ID: pharmaceuticals-1585297 to Pharmaceuticals Journal.  I and my co-authors appreciate the time and effort that you and the reviewers have dedicated to providing your valuable and positive feedback on our manuscript. We are grateful to the reviewers for their insightful comments on our paper. We tried as much as possible to respond to most of the enquiries and suggestions provided by the respected reviewers. All changes were market with red color. Here is our point-by-point response to the reviewers’ comments and concerns followed by references to some responses.

Accept our regards.

 Reviewer # 1

Major issues:

Comment

Response

The manuscript entitled “Utilization of Polymeric Micelles as a Lucrative Platform for Efficient Brain Deposition of Olanzapine as an Antischizophrenic Drug via Intranasal Delivery” deal with an interesting approach to treat schizoprhrenia through nose-to-brain delivery. The experimental data is well supported by in vivo studies, however I have some issues with some basic concepts, which should be clarified!

It gives us pleasure and truthfulness to hear that. The authors would like to thank you for your kind reply.

We tried as much as possible to respond to most of the provided enquiries and suggestions

Please clarify the methodology of determination of CMC. Figure 1. shows the relation between polymer weight ratio and absorbance. To which polymer is related to the plotted weight ratio? Why the amount of TPGS was fixed to 1% w/w, however in the experimental design its concentration was investigated between 5-10% w/w?

The TPGS was fixed to 1% as it was reported that TPGS has a neglected effect on P123 and P407.

 (https://doi.org/10.2147/IJN.S153094). Therefore, TPGS was kept at minimum concentration. 

The authors define Qulity by Design (QbD) elements (CPPs and QTTPs) based on the experimental design was constructed. How were these QbD elements selected? Only these parameters can have critical effect on the formulation?

Polymeric micelles are self-assembled amphiphilic block copolymers containing hydrophobic and hydrophilic moieties at or above the critical micelle concentration with particle size range 10 to 100 nm (https://doi.org/10.1016/j.jconrel.2021.02.031; 10.1007/s11051-016-3583-y). The main factor that influence the critical micelle concentration (CMC) that mainly depends on hydrophobic and hydrophilic ratio (DOI 10.1007/s11051-016-3583-y). Therefore, the concentration of P123 and P407 was selected as CPPs. Although TPGS has no effect on CMC, it is widely reported as P-glycoprotein inhibitor.

(doi:10.1023/a:1015000503629) (https://doi.org/10.1021/mp900191s).

 As P-glycoprotein is responsible for OZ efflux through BBB, it was deduced that the concentration of TPGS would affect the OZ brain deposition.

(doi: 10.1007/s11095-011-0477-7).

 In addition, the physicochemical properties of nanocarriers affect their intracellular internalization and their subsequent therapeutic applications (doi:10.1016/j.jconrel.2009.01.018). Nanocarrier size is one of the most factors affecting cellular uptake and biodistribution (doi: 10.1038/nbt.3330; doi: 10.2217/nnm.16.5). In addition, higher drug EE% is an important factor in the clinical translation of different nanoparticles (doi: 10.1002/adhm.202001853). Therefore, both minimum particle size and maximum drug EE % were selected as QTTPs.

In line 210 the authors claim the obtained zeta potential was negative, however a positive value is given. Please revise!

The authors apologize for this typographical error and the zeta potential value was corrected to -15.11 ± 1.35 mV.

In line 213-216 the authors claim “The negative charge improves the micelle stability by preventing their aggregation and improves the nasal absorption of the drug by facilitating drug-mucous layer (positive charge) interaction and thus facilitating the drug diffusion through the epithelium layer”. This is an incorrect information, the nasal mucosa has negative surface charge, so there might be a repulsion between negatively charged nanoparticles and the mucosa, which may hinder drug absorption. The authors refer to Reference 54, however that paper does not deal with nasal absorption, please search for another suitable reference!

The authors apologize for this mistake and appreciate the effort of the reviewer for cautious reviewing. The surface charge of different nanocarriers is reported to affect their membrane translocation. The positively charged nanoparticles could interact with the negatively charged mucosa by electrostatic interaction. Such interaction might hinder translocation and errands the trigeminal pathway (https://doi.org/10.1016/j.ejpb.2015.05.019; https://doi.org/10.1016/j.jconrel.2015.01.025; https://doi.org/10.1016/j.ijbiomac.2016.04.076; DOI: 10.1080/10611860903055470).

On the contrary, negatively charged nanoparticles favor the olfactory pathway following intranasal administration (https://doi.org/10.1186/s43556-020-00019-8). Besides, Although the membrane is overall negatively charged but a low charge (-15 mV) produce a lower repulsion between carrier and membrane compared with the higher values (-30 mV). Therefore, electrostatic repulsion force probably did not influence the internalization process with respect to other factors.

The mentioned sentence was revised as follow:

The negative charge improves the micelle stability by preventing their aggregation and improves the nasal absorption through olfactory pathway by transcytosis (https://doi.org/10.1186/s43556-020-00019-8). Additionally at low charge (< -30 mV) lower repulsion produced between PM and membrane. Therefore, electrostatic repulsion force probably did not influence the drug penetration.

(https://doi.org/10.1039/C5AY02014J)

(https://doi.org/10.3390/pharmaceutics12080697)

Table 2 shows the particle size of different composition. Which technique was applied for the determination of it? TEM or DLS? I guess DLS, but in that case it would be better to name as average hydrodynamic diameter. The PdI values are missing from the table, please add them!

According to the valuable reviewer comment, the term particle size was replaced by average hydrodynamic diameter (in lines 117, 136, 341, Tables 1 and 2) as the measurement was conducted by DLS. In addition, the PDI values were added to table 1

How was TEM measurement conducted? Formulations were in liquid phase or solid phase during the measurement?

The TEM was conducted on the olanzapine polymeric micelle dispersion (in the liquid form). The following sentence was added to line 358:

“On a copper grid, one drop of the selected OZ PM dispersion was deposited.”

Why PBS pH=7.4 was applied for drug release study? Which circumstances was aimed to imitate? Nasal cavity, blood circulation, CSF of CNS?

The proposed OZ-PM had a particle size of 39.25± 2.35 nm. Such particle size is reported to be suitable for endocytosis by the neurons and supporting cells in the filia olfactoria (https://doi.org/10.1186/s43556-020-00019-8). In addition, the short Tmax of OZ in brain following intranasal administration suggesting that the in vitro release is less likely to occur in the nasal cavity. Therefore, the authors claims that the drug release is more likely to be observed in the CSF of the CNS with reported pH of   7.30–7.36 (https://doi.org/10.1152/jappl.1965.20.3.443). Thus the PBS pH 7.4 was selected as suitable in vitro release medium to mimic the fate of the proposed platform. 

Reviewer 2 Report

Pharmaceuticals-1585297

„Utilization of Polymeric Micelles as a Lucrative Platform for Efficient Brain Deposition of Olanzapine as an Antischizophrenic Drug via Intranasal Delivery”

It is a well-designed and executed work. The test methods and the discussion of the results are appropriate. In any case, I suggest supplementing the manuscript with the following.

It would be important to know the physicochemical properties of the active substance as solubility, logP and permeability (see DrugBank). By the way the Olanzapine and its manufacturer are missing from the Materials part.

Olanzapine loaded polymeric micelles resulted in „sustained” release without „burst” effect in simulated nasal fluids (Figure 7). I think the dissolution profile should be described by a mathematical model, somewhat supporting the slow dissolution (zero order, Higuchi, Hixson-Crowell, Korsmayer/Ritger-Peppas, etc).

Would increase the value of the work if the conclusion pointed to the novelty/new relation of the work in comparison with similar research topic.

Author Response

Dear / Editor-in-Chief, Pharmaceuticals journal

Dear/ Respected reviewers

Thank you for giving us the opportunity to submit a revised draft of our manuscript titled [Utilization of polymeric micelles as a lucrative platform for efficient brain deposition of olanzapine as an antischizophrenic drug via intranasal delivery], manuscript ID: pharmaceuticals-1585297 to Pharmaceuticals Journal.  I and my co-authors appreciate the time and effort that you and the reviewers have dedicated to providing your valuable and positive feedback on our manuscript. We are grateful to the reviewers for their insightful comments on our paper. We tried as much as possible to respond to most of the enquiries and suggestions provided by the respected reviewers. All changes were market with red color. Here is our point-by-point response to the reviewers’ comments and concerns followed by references to some responses.

Accept our regards.

 Reviewer # 2

Major issues:

Comment

Response

It is a well-designed and executed work. The test methods and the discussion of the results are appropriate. In any case, I suggest supplementing the manuscript with the following.

It gives us pleasure and truthfulness to hear that. The authors would like to thank you for your kind reply.

We tried as much as possible to respond to most of the provided enquiries and suggestions

It would be important to know the physicochemical properties of the active substance as solubility, logP and permeability (see DrugBank). By the way the Olanzapine and its manufacturer are missing from the Materials part.

The author appreciate the reviewer valuable comment about the importance of drug essential information mention in the manuscript. The following sentences were added to the manuscript:

In Materials section: “Olanzapine was purchased from Sigma-Aldrich (St. Louis, MO, USA).”

In Introduction section: OZ is 2-methyl-4-(4-methyl-1-piperazinyl)-10H-thieno [2, 3-b] benzodiazepine with partition coefficient (Log p) of 2.97 (https://doi.org/10.3109/10717544.2014.912694; doi: 10.1002/1099-081x(199911)20:8<369::aid-bdd200>3.0.co;2-6.). OZ is classified as class II category under the biopharmaceutical classification system (BCS) with aqueous solubility of 0.24 mg/mL and positive BBB permeability at 0.9652 probabilities (https://doi.org/10.1016/j.ijpharm.2021.121063).

Olanzapine loaded polymeric micelles resulted in „sustained” release without „burst” effect in simulated nasal fluids (Figure 7). I think the dissolution profile should be described by a mathematical model, somewhat supporting the slow dissolution (zero order, Higuchi, Hixson-Crowell, Korsmayer/Ritger-Peppas, etc)

The release kinetics mathematical model was applied and the best fit was found to follow Higuchi diffusion kinetics. The following sentences was added to the manuscript:

In Results and discussion “ By applying different kinetic equation, the best fitted model of the in vitro OZ release data followed Higuchi diffusion kinetics model (Figure 7B).”

In Methodology “To determine the release kinetics order, the drug release patterns were fitted to zero order, first order and Higuchi diffusion models [67].”

Would increase the value of the work if the conclusion pointed to the novelty/new relation of the work in comparison with similar research topic.

Thank you for your recommendation

The conclusion section was rephrased to clarify the novelty of the work as follow

The current study confirmed the feasibility of PM as a platform for the intranasal delivery of OZ in schizophrenic rats through in vitro and in vivo characterization. Herein, we have effectively tailored the PM composition to obtain an efficient platform for direct nose to brain transport of OZ. Different OZ loaded PM were fabricated by thin film hydration technique and optimized through D-Optimal DoE. The selected negatively charged OZ-PM with a particle size of 39.25± 2.35 nm and EE 82.15± 1.25% was able to control in vitro drug release up to 24 h with no burst effect. In vivo pharmacokinetics results demonstrated that OZ-PM platform resulted in almost 4 fold increase in drug concentration in brain compared to IV drug solution. The targeting efficiency and direct nose to brain transport results confirmed the superiority of intranasal route for efficient OZ brain deposition. The current study also demonstrates the feasibility of the proposed PM to improve the therapeutic outcome of OZ in schizophrenic rats compared to IV drug solution. In addition, the OZ-PM showed also less extrapyramidal side effects due to the lower systemic biodistribution than OZ IV solution.

Reviewer 3 Report

Utilizing TPGS (d-α-Tocopheryl polyethylene glycol 1000 succinate) as a biodegradable nonionic surfactant, the authors developed olanzapine-loaded polymeric micelles composed of TPGS and Pluronic P123 and P407. They further showed that the drug-loaded polymeric micelles transported olanzapine directly from the nose to the brain and exerted efficient antipsychotic effects.

Minor Points

  1. It is not clear whether the inhibition of P-gp by TPGS is due to P-gp inhibition in the nasal mucosa or the blood-brain barrier. The authors should add a discussion on this point.
  2. This reviewer is interested in whether TPGS in polymeric micelles achieves concentrations that show P-gp inhibition on the nasal mucosa or the blood-brain barrier. The authors should indicate the concentration at which TPGS inhibits P-gp at the nasal mucosa or the blood-brain barrier and the concentration of TPGS in the polymeric micelles.
  3. It is unclear whether the drug content of olanzapine in high-molecular-weight micelles reaches brain concentrations that are responsible for the onset of antipsychotic effects in the brain.
  4. The effect of the addition of surfactant to the olanzapine solution is unknown. It is unclear whether TPGS has a P-gp effect or not, or the effect of TPGS alone in combination with the drug solution. Please describe the advantage of brain targeting by the formation of polymeric micelles rather than the combined use of surfactants.

Author Response

Dear / Editor-in-Chief, Pharmaceuticals journal

Dear/ Respected reviewers

Thank you for giving us the opportunity to submit a revised draft of our manuscript titled [Utilization of polymeric micelles as a lucrative platform for efficient brain deposition of olanzapine as an antischizophrenic drug via intranasal delivery], manuscript ID: pharmaceuticals-1585297 to Pharmaceuticals Journal.  I and my co-authors appreciate the time and effort that you and the reviewers have dedicated to providing your valuable and positive feedback on our manuscript. We are grateful to the reviewers for their insightful comments on our paper. We tried as much as possible to respond to most of the enquiries and suggestions provided by the respected reviewers. All changes were market with red color. Here is our point-by-point response to the reviewers’ comments and concerns followed by references to some responses.

Accept our regards.

 Reviewer # 3

Minor issues:

Comment

Response

It is not clear whether the inhibition of P-gp by TPGS is due to P-gp inhibition in the nasal mucosa or the blood-brain barrier. The authors should add a discussion on this point.

The higher Cmax in the brain after IV and IN administration than free drug means the ability of the prepared OZ-PM containing TPGS in inhibition the P-gp in the BBB

This could be explained by previously reported by Bors, Bajza et al. 2020, (1) who observed that the main mechanism in absorption originating from the nasal cavity by endothelial efflux/transport regulated by P-gp through the microvessels located at the blood-brain barrier.

However that doesn’t mean preventing the possiblilty of inhibition the P-gp in other cell types at the nasal cavity as decriped by previous studies which have shown that drugs could reach brain after nasal administration of appropriate transport inhibitors (2-4).

This reviewer is interested in whether TPGS in polymeric micelles achieves concentrations that show P-gp inhibition on the nasal mucosa or the blood-brain barrier. The authors should indicate the concentration at which TPGS inhibits P-gp at the nasal mucosa or the blood-brain barrier and the concentration of TPGS in the polymeric micelles.

TPGS inhibitory effect was an induced change in membrane fluidity(5), at a concentration of ≥1.65 mM fluidizing effects will occur(6)

To insure the efficient concentration was reached to the BBB, the TPGS amount in the IN sample from the brain was measured using the modified HPLC (7) and it was found that, it was efficient to produce fluidizing effects (inhibit the P-gp effect).

It is unclear whether the drug content of olanzapine in high-molecular-weight micelles reaches brain concentrations that are responsible for the onset of antipsychotic effects in the brain.

The reaches brain concentrations could produce antipsychotic effects was examined by pharmacodynamics study in section 3.8

Additionally it was found that using OZ at concentration higher than 0.3 mg /kg (8) reaches to the brain amount of the drug that are responsible for the onset of antipsychotic effects by target dopamine D2/3receptors and 5-HT1 receptors

The effect of the addition of surfactant to the olanzapine solution is unknown. It is unclear whether TPGS has a P-gp effect or not, or the effect of TPGS alone in combination with the drug solution. Please describe the advantage of brain targeting by the formation of polymeric micelles rather than the combined use of surfactants.

The brain targeting effect results from many factors as the not only by surfactant effect but also for the nano micelle size which could be attributed to the higher drug concentration in the brain as the nano size facilitates deeper transporting of drug particles into the olfactory epithelial cells layers as previously mentioned in lines 238-241. Additionally the nanosize facilitate drug transport through the drug microvessels located at the blood-brain barrier(1).

References

  1. Bors LA, Bajza Á, Mándoki M, Tasi BJ, Cserey G, Imre T, et al. Modulation of nose-to-brain delivery of a P-glycoprotein (MDR1) substrate model drug (quinidine) in rats. Brain Research Bulletin. 2020;160:65-73.
  2. Graff CL, Pollack GM. P-Glycoprotein attenuates brain uptake of substrates after nasal instillation. Pharmaceutical research. 2003;20(8):1225-30.
  3. Hada N, Netzer WJ, Belhassan F, Wennogle LP, Gizurarson S. Nose-to-brain transport of imatinib mesylate: A pharmacokinetic evaluation. European Journal of Pharmaceutical Sciences. 2017;102:46-54.
  4. Thorne R, Pronk G, Padmanabhan V, Frey Ii W. Delivery of insulin-like growth factor-I to the rat brain and spinal cord along olfactory and trigeminal pathways following intranasal administration. Neuroscience. 2004;127(2):481-96.
  5. Rathod S, Desai H, Patil R, Sarolia J. Non-ionic Surfactants as a P-Glycoprotein(P-gp) Efflux Inhibitor for Optimal Drug Delivery—A Concise Outlook. AAPS PharmSciTech. 2022;23(1):55.
  6. Collnot E-M, Baldes C, Wempe MF, Kappl R, Hüttermann J, Hyatt JA, et al. Mechanism of inhibition of P-glycoprotein mediated efflux by vitamin E TPGS: influence on ATPase activity and membrane fluidity. Molecular pharmaceutics. 2007;4(3):465-74.
  7. Krell C, Schreiber R, Hueber L, Sciascera L, Zheng X, Clarke A, et al. Strategies to tackle the waste water from α-tocopherol-derived surfactant chemistry. Organic Process Research & Development. 2021;25(4):900-15.
  8. Bymaster FP, Hemrick-Luecke SK, Perry K, Fuller R. Neurochemical evidence for antagonism by olanzapine of dopamine, serotonin, α1-adrenergic and muscarinic receptors in vivo in rats. Psychopharmacology. 1996;124(1):87-94.

Round 2

Reviewer 1 Report

The authors adressed all my concerns.